# Review of Photothermal Technique for Thermal Measurement of Micro-/Nanomaterials

**DOI:** 10.3390/nano12111884

**Published:** 2022-05-31

**Authors:** Jianjun Zhou, Shen Xu, Jing Liu

**Affiliations:** 1School of Mechanical and Automotive Engineering, Shanghai University of Engineering Science, Shanghai 201620, China; jjzhousues@163.com; 2College of New Materials and New Energies, Shenzhen Technology University, Shenzhen 518116, China; liujing@sztu.edu.cn

**Keywords:** photothermal technique, thermal properties, nanostructure characterization, thermal conductivity, specific heat, thermal effusivity

## Abstract

The extremely small size of micro-/nanomaterials limits the application of conventional thermal measurement methods using a contact heating source or probing sensor. Therefore, non-contact thermal measurement methods are preferable in micro-/nanoscale thermal characterization. In this review, one of the non-contact thermal measurement methods, photothermal (PT) technique based on thermal radiation, is introduced. When subjected to laser heating with controllable modulation frequencies, surface thermal radiation carries fruitful information for thermal property determination. As thermal properties are closely related to the internal structure of materials, for micro-/nanomaterials, PT technique can measure not only thermal properties but also features in the micro-/nanostructure. Practical applications of PT technique in the thermal measurement of micro-/nanomaterials are then reviewed, including special wall-structure investigation in multiwall carbon nanotubes, porosity determination in nanomaterial assemblies, and the observation of amorphous/crystalline structure transformation in proteins in heat treatment. Furthermore, the limitations and future application extensions are discussed.

## 1. Introduction

With reductions in size to the micro-/nanometer level, temperature probing and thermal measurement have become difficult to conduct using traditional contact-based methods and equipment (thermal couples and thermistor, etc.). Non-contact thermal measurement methods have thus become prevalent for the thermal characterization of micro-/nanomaterials [1,2]. Non-contact methods mainly benefit from the laser heating source and thermally induced phenomena, which can be detected from a distance [3,4,5,6,7,8,9,10,11]. Based on the features of the phenomena, the widely adopted non-contact thermal methods are typically divided into three types: time-domain techniques, frequency-domain techniques, and spectroscopy [1].

Among these techniques, time-domain thermoreflectance (TDTR) [7,12,13] and frequency-domain thermoreflectance (FDTR) [14,15] detect the temperature rise by sensing changes in the surface optical properties in the time and frequency domains. They utilize an ultrafast heating pulse to generate a nanometer-level thermal penetration depth and thus have a good ability to measure the in-plane and out-of-plane thermal conductivity for thin films and bulks. The obvious drawbacks of these two methods are that they require smooth surfaces and post-processing. The photoacoustic (PA) method is a frequency-domain method that measures the surface temperature by detecting the sound waves produced by the work done by the periodical thermal expansion of the heated surface [3]. Avoiding the mechanical piston effect induced by the thermal expansion of the heated surface, the modulation frequency of the heating laser is limited so that it is lower than the order of 10 kHz [9,11,16]. The PA method can work well with micrometer-thick films and bulks due to the long thermal penetration depth at a lower frequency [9]. Furthermore, as it is limited by the microphone, the PA method is usually deployed at room temperature [3,9,11]. The laser flash method [17,18] involves heating a suspended material on the front side and analyzing the transient temperature rise in the time domain from the back based on the thermal radiation. Compared with the PA method, the laser flash method can be used with a wide temperature range from −125 to 2800 °C [1]. However, this method has thickness limitations with regard to samples [19,20]. More recently, Raman-based thermal methods have become popular due to its feature of being material-specific. The steady-state Raman method has a simple physical mechanism for thermal characterization [21,22], while the transient Raman method offers high accuracy in measurement results [8,23,24,25,26,27,28,29,30,31]. It is notable that the temperature probing depth with the Raman method is usually tens of nanometers. It is less often used to measure the thermal properties of thick films or bulks. More comprehensive reviews of general photothermal technologies can be found in [1,2].

In this paper, a short review is provided for the photothermal (PT) technique based on thermal radiation established in Wang’s lab [32,33]. This PT technique stems from PA technology. However, in contrast to PA technology, PT technique acquires the frequency-domain thermal radiation instead of sound waves, which can reduce the complexity of the measurement setup and widen the temperature range for thermal measurement, as the microphone is no longer necessary. Furthermore, it has no frequency limitation and, thus, can theoretically measure samples with thicknesses ranging from nanoscale to bulk. The PT method also has a low requirement for smooth surfaces when compared to thermoreflectance methods because it detects thermal radiation rather than reflections. In Section 2 and Section 3, the theory and a typical experimental setup for PT technique are introduced. Section 4 discusses the application of PT technique in the thermal characterization of micro-/nanomaterials, especially the measurement of thermophysical properties and structure probing. Furthermore, considerations for the thermal measurement of micro-/nanomaterials using PT technique are also discussed.

## 2. PT Theory for Thermal Property Measurements

The PT technique developed in Wang’s lab employs a periodically modulated laser source to heat a solid surface. In each period, the surface temperature immediately rises after heating is applied. The speed and intensity of the thermal response of the surface are strongly dependent on the thermal properties of the materials under the surface. Thermal radiation due to surface temperature rise carries important information regarding the thermal properties of the materials and structures beneath, both for homogeneous and multilayered structures.

### 2.1. Physical Model Derivation

PT technique stems from the physical model of PA technology proposed by Rosencwaig et al. [3], which is a one-dimensional cross-plane heat conduction model in a multilayered structure, as shown in Figure 1a. The model requires that the size of the heating source be much larger than heat diffusion length in each layer, so that the in-plane heat conduction can be safely neglected and the generated heat conducts one-dimensionally along the cross-plane direction. Furthermore, the surface temperature rise should be moderate, and the heat loss through thermal convection and radiation is reasonably negligible. Hence, the governing equation of 1D cross-plane heat conduction under periodical heating is
(1)$$\frac{\partial^{2}\theta_{i}}{\partial x^{2}}=\frac{1}{\alpha_{i}}\frac{\partial \theta_{i}}{\partial t}-\frac{\beta_{i}I_{0}}{2\kappa_{i}}\exp\left(\sum_{m=i+1}^{N}-\beta_{m}L_{m}\right)\times e^{\beta_{i}(x-l_{i})}(1+e^{j\omega t}),$$

The subscript *i* means that the physical properties are for a certain layer *i*; therefore, *θ_i_ = T_i_ − T_amb_* is the temperature rise of layer *i* and *T_amb_* is the ambient temperature. *I*_0_ is the incident laser power and *ω* is the angular frequency (2*πf*) corresponding to the modulation frequency *f*. *α**_i_*, *κ_i_,* and *β_i_* are the thermal diffusivity, thermal conductivity, and optical absorption coefficient for layer *i*. *L_i_ = l_i_ − l_i−_*_1_ is the thickness of layer *i*, where *l_i_* is the surface location of layer *i* on the *x* axis in Figure 1. *j* is $\sqrt{-1}$. A detailed derivation of Equation (1) is provided in [9]. 

The resultant surface temperature rise *θ_i_* for layer *i* gradually increases from zero to a new steady state with fluctuations. Thus, *θ_i_* can be divided into three components: the transient component, *θ_i,t_*; the steady DC component, ${\bar{\theta }}_{i,s}$; and the steady AC component, ${\tilde{\theta }}_{i,s}$. *θ_i,t_* represents the initial temperature rise immediately after the laser heating is applied. When the sample reaches the steady state, ${\bar{\theta }}_{i,s}$ indicates the steady state temperature, while ${\tilde{\theta }}_{i,s}$ is the fluctuation in temperature due to the modulated heating source. ${\tilde{\theta }}_{i,s}$ is easily determined by a lock-in amplifier at a set modulation frequency. It has an explicit expression as follows:(2)$${\tilde{\theta }}_{i,s}=[A_{i}e^{\sigma_{i}(x-l_{i})}+B_{i}e^{-\sigma_{i}(x-l_{i})}-E_{i}e^{\beta_{i}(x-l_{i})}]e^{j\omega t},$$
where $E_{i}=G_{i}/(\beta_{i}^{2}-\sigma_{i}^{2})$ with $G_{i}=\beta_{i}I_{0}/\left(2k_{i}\right)\exp\left(-\sum_{m=i+1}^{N}\beta_{m}L_{m}\right)$, and for *I* < *N*, $G_{N}=\beta_{N}I_{0}/2k_{N}$, and *G_N_*_+1_ = 0. *σ_I_* is (1 + *j*)·*a_i_*, where $a_{i}=1/\mu_{i}$ is the thermal diffusion coefficient and $\mu_{i}=\sqrt{\alpha_{i}/\pi f}$ is the thermal diffusion length. 

*A_i_* and *B_i_* are important coefficients derived from the interfacial transmission matrix of heating *U* and the absorption matrix of light *V*:(3)$$\left[\begin{array}{c} A_{i}\\ B_{i} \end{array}\right]=U_{i}\left[\begin{array}{c} A_{i+1}\\ B_{i+1} \end{array}\right]+V_{i}\left[\begin{array}{c} E_{i}\\ E_{i+1} \end{array}\right],$$
where *U_i_* and *V_i_* from layer *i* + 1 to *I* are
(4)$$U_{i}=\frac{1}{2}\left[\begin{array}{cc} u_{11,i} & u_{12,i}\\ u_{21,i} & u_{22,i} \end{array}\right];\;V_{i}=\frac{1}{2}\left[\begin{array}{cc} v_{11,i} & v_{12,i}\\ v_{21,i} & v_{22,i} \end{array}\right],$$
where
(5)$$u_{1n,i}=\left(1\pm k_{i+1}\sigma_{i+1}/k_{i}\sigma_{i}\mp k_{i+1}\sigma_{i+1}R_{i,i+1}\right)\times \exp\left(\mp \sigma_{i+1}L_{i+1}\right),\;n=1,\;2,$$
(6)$$u_{2n,i}=\left(1\pm k_{i+1}\sigma_{i+1}/k_{i}\sigma_{i}\mp k_{i+1}\sigma_{i+1}R_{i,i+1}\right)\times \exp\left(\mp \sigma_{i+1}L_{i+1}\right),\;n=1,\;2,$$
(7)$$v_{n1,i}=1\mp \beta_{i}/\sigma_{i},\;n=1,\;2,$$
and
(8)$$v_{n2,i}=\left(-1\mp k_{i+1}\beta_{i+1}/k_{i}\sigma_{i}\mp k_{i+1}\beta_{i+1}R_{i,i+1}\right)\times \exp\left(-\beta_{i+1}L_{i+1}\right),\;n=1,\;2.$$

*R_i_*_,_*_i_*_+1_ is the thermal contact resistance between layer *i* and (*i* + 1). It is noticeable that the thermal and optical properties of the materials in the multilayered structure are all included in Equations (5)–(8). Thus, the thermal properties are closely related to the temperature rise ${\tilde{\theta }}_{i,s}$ of the layer *i*. 

Under the assumption that the front air layer and back substrate are thermally thick—that is, $\left|\sigma_{0}L_{0}\right|\gg 1$ and $\left|\sigma_{N+1}L_{N+1}\right|\gg 1$—*A_N_*_+1_ and *B*_0_ are equal to zero. Then, applying the interfacial condition between layer *i* and (*i* + 1),
(9)$$k_{i}\frac{\partial {\tilde{\theta }}_{i,s}}{\partial x}-k_{i+1}\frac{\partial {\tilde{\theta }}_{i+1,s}}{\partial x}=0$$
(10)$$\mathrm{and}\;k_{i}\frac{\partial {\tilde{\theta }}_{i,s}}{\partial x}+\frac{1}{R_{i,i+1}}\left({\tilde{\theta }}_{i,s}-{\tilde{\theta }}_{i+1,s}\right)=0,$$
the solved *A_i_* and *B_i_* are
(11)$$\left[\begin{array}{c} A_{i}\\ B_{i} \end{array}\right]=(\prod_{m=i}^{N}U_{m})\left[\begin{array}{c} 0\\ B_{N+1} \end{array}\right]+\sum_{m=i}^{N}\left(\prod_{k=i}^{m-1}U_{k}\right)V_{m}\left[\begin{array}{c} E_{m}\\ E_{m+1} \end{array}\right]$$
(12)$$B_{N+1}=-\frac{\left[\begin{array}{cc} 0 & 1 \end{array}\right]\sum_{m=0}^{N}(\prod_{i=0}^{m-1}U_{i})V_{m}\left[\begin{array}{c} E_{m}\\ E_{m+1} \end{array}\right]}{\left[\begin{array}{cc} 0 & 1 \end{array}\right](\prod_{i=0}^{m-1}U_{i})\left[\begin{array}{c} 0\\ 1 \end{array}\right]}$$

By substituting *A_i_*, *B_i_*, and *E_i_* into Equation (2), we can obtain the temperature distribution in any layer of interest. This greatly increases the flexibility of the PT method. For the purpose of non-contact measurement, an infrared detector is usually employed to gather the surface radiation from either the front or back. 

### 2.2. Phase Shift and Amplitude

The AC temperature rise component ${\tilde{\theta }}_{i,s}$ has two critical properties, amplitude and phase. Compared with the original periodical heating source, the occurrence of temperature rises and thermal radiation is delayed by the heat conduction inside the multilayered structure. Correspondingly, the phase of the radiation is slower than the phase of the heating source, and the difference (phase shift) between these two can be deducted to be *Arg*(*B_N_*_+1_) − *π*/4. According to Equation (12), the thermal properties are included in *B_N_*_+1_. Measurement based on the phase shift—the phase shift method—can be used to accurately evaluate the thermal properties of a specific layer and the interfacial thermal conductance in the multilayered structure due to the high sensitivity of ~0.1° [32]. However, for bulk materials with a smooth surface, it becomes a constant of −45° [16]. In contrast, the amplitude of thermal radiation is proportional to the temperature rise. Since the thermal diffusion depth is different with different modulation frequencies, the amplitude of the thermal radiation changes against the frequency. Alternatively to the phase shift method, measurement based on the amplitude is able to determine the thermal properties of the bulk materials. 

## 3. Experimental Implementation of PT Method for Thermal Property Measurements

### 3.1. Experimental Setup

A typical measurement setup using the PT technique developed in Wang’s lab [11] is shown in Figure 1b. The function generator-modulated diode laser (at a visible wavelength) is focused on a sample surface by using a focal lens to heat the sample. Then, a pair of off-axis parabolic mirrors collect the raised thermal radiation resulting from the temperature rise and send it to an infrared (IR) detector. Along with the radiation collection, the diffuse reflection of the incident laser from the surface is also gathered. Though the IR detector is much less sensitive to the visible wavelength, the reflection is still much stronger than the radiation. Thus, an IR window (germanium (Ge) window in Figure 1b) is placed in front of the detector to eliminate the visible diffuse reflection and let only the thermal radiation enter into the detector. The radiation-converted voltage signal is then intensified in a preamplifier and finally analyzed in a lock-in amplifier to extract the phase shift and the amplitude when compared with the reference signal from the function generator. 

When using PT technique, given that the unexpected, complex photon–electron–phonon process may occur under laser irradiation in the sample, especially for semiconductors, a metal coating (usually gold, aluminum, etc.) is applied to the sample surface to act as a well-defined energy absorber and heater. The coating is optically thick and can totally absorb the incidence. At the same time, it is thermally thin and has a negligible effect on heat conduction (phase shift). It is physically understandable that the heat diffusion length/depth in the multilayered structure should be controllable by changing the modulating frequency. The selection of the modulating frequency of the heating laser needs to be evaluated in advance because the sample layer in the multilayered structure should be involved in the heat conduction. 

### 3.2. System Calibration

The raw data recorded by the lock-in amplifier—the phase shift *ϕ_raw_* and amplitude *A_raw_*—are not available for direct analysis because the measurement system induces additional errors in the raw data (the phase shift and amplitude). For example, the optical path and electric devices involved raise an additional time delay in the phase shift, and the fluctuations in the laser power, as well as the optical path, cause unexpected variations in amplitude. Thus, calibration of the measurement system is necessary to exclude these effects from the raw data. The diffuse reflection of the incident laser is measured to calibrate the measurement system, since it passes through the same path as the thermal radiation does. The calibrated phase shift *ϕ_cal_* and amplitude *A_cal_* are shown in Figure 2. To calibrate *ϕ_raw_*, the absolute phase shift due to heat conduction is quickly determined as *ϕ_nor_* = *ϕ_raw_* − *ϕ_cal_*. For the amplitude, it is more complicated. The amplitude is affected by not only the laser power and attenuation in the optical path but also the modulation frequency *f*. Xu et al. proposed the equation $A_{nor}=A_{raw}\cdot \sqrt{f}/A_{cal}$ to normalize *A_raw_* and exclude all the possible errors induced by the system, detailed in [33]. The calibrated *A_nor_* is approximated to *ζ*/*e_t_*, where *ζ* is a system-related constant and the effusivity $e_{t}=\sqrt{\kappa \rho c_{p}}$. It directly correlates *A_nor_* with the thermal properties. After calibration, *ϕ_nor_* and *A_nor_* can be used to determine the thermal properties of the materials of interest.

### 3.3. Uncertainty

During the data measurement, different heat conduction processes occur when the modulation frequency is changed. Then, the thermal properties can be determined by fitting the temperature variation against the frequencies. Wang et al. found that the uncertainty is related to the ratio of the thermal diffusion length *μ_i_* to the layer thickness *L_i_* [11]. Based on their SiO_2_/Si sample, the numerical uncertainty for the phase shift method was around ±5%when *μ_i_* and *L_i_* were in the same order, and it increased to ±15% when *μ_i_*/*L_i_* was around 100. For the case in which *μ_i_*/*L_i_* was less than 0.15, the thermal energy would not diffuse across the SiO_2_ layer. The phase shift method would not be suitable for this case, while the amplitude method showed a ±10% uncertainty. Xu et al. studied a similar SiO_2_/Si sample and achieved an experimental uncertainty of 5% based on the phase shift method and 10% with the amplitude method [33]. For thermal contact resistance measurement, the experimental sensitivity and uncertainty were limited by the uncertainty in the thermal conductivity discussed above. The sensible limit was reported to be 10^−8^ m^2^K/W [11] for the phase shift method and 10^−7^ m^2^K/W [33] for the amplitude method. Since the thermal contact resistance fell in the range of 10^−9^–10^−7^ m^2^K/W, the amplitude method was not sensitive to the thermal contact resistance. It could thus accurately measure the thermal conductivity without knowledge of the interface.

## 4. PT Measurement of Nanomaterials

When the characteristic lengths of materials are reduced to the micro-/nanoscale, the thermal properties also significantly decrease due to the size effect. Based on this fact, investigation of thermal properties can be an efficient supplementary way to characterize the micro-/nanostructure in addition to the most commonly used micro-/nanoscale imaging. 

### 4.1. Nanostructure Analysis through Thermal Characterization

Wang et al. [32,34] adopted the PT method and studied the axial thermal conductivity of multiwall carbon nanotubes (CNTs) prepared using plasma-enhanced chemical vapor deposition (PECVD). The CNT sample for PT measurement was composed of three layers. As in Figure 3a, the layers from top to bottom were a thin silicon wafer (14 μm thick), a layer of chromium (Cr, 70 nm thick), and the layer of vertically aligned CNTs. Between the Cr layer and CNTs, there was a thin nickel (Ni) film of a negligible thickness, which offered seeds for CNTs’ growth. The Si wafer was transparent to the incident laser wavelength (1064 nm) and thermal radiation. Therefore, the incident laser heated the Cr film, the partial generated heat was conducted along the axial direction of the CNTs, and the radiation from the Cr surface was analyzed to obtain the axial thermal conductivity of the CNTs. The resultant thermal conductivity of 27.3 W/m·K was dramatically lower than the theoretical thermal conductivity of 1600–6600 W/m·K for single-wall CNTs, where phonons can conduct heat in a perfect wall plane. Combined with the TEM result for the CNTs, it was found that the special structure of the Ni seeds led to the CNTs’ walls being tilted with respect to the tube axis, as shown in Figure 3b. This unexpected structure raised a large number of boundaries along the axial direction and thus reduced the axial thermal conductivity of the CNTs. Here, though the PT method measured the CNTs’ axial thermal conductivity as a bulk, the result greatly helped interpret the special growth mechanism of the CNTs.

### 4.2. Porosity Determination in Nanostructures

For loosely assembled nanoparticles, porosity is an important parameter demonstrating the quality of the assembly, but it is hard to measure by mapping only the surface. Pores and cavities in the nanostructure generate additional defects and boundaries and then reduce the thermal properties of the assembled nanostructure. Based on this mechanism, Chen et al. [35] measured the effective thermal conductivity and volumetric heat capacity of a hydrogenated vanadium-doped magnesium (V-doped Mg) porous nanostructure using PT technique. Under the effect of cavities on the V-doped Mg composite (MgH_2_ was the main component), the effective volumetric heat capacity was apparently lower than that of the MgH_2_ bulk counterpart. The volumetric heat capacity ratio of nanostructure to bulk helped further reveal the porosity level *φ* of the nanostructure. The determined porosity level was validated through SEM observation. The porosity level φ was estimated to be 25–42% from SEM, and the φ calculated from the PT results was 9.0–39.4%, with an upper limit falling into the range of the SEM observation. It should be noted that the SEM observation scale (microscale) was much smaller than that probed with PT technique (~millimeter scale). Thus, the PT-determined porosity level is more applicable when the size of nanostructured assemblies reaches the macroscale. The intrinsic thermal conductivity of the solid part of the porous nanostructure was then determined to be ~3.5 W/m·K, while it had been ~1.9 W/m·K before excluding the effect of the cavities. PT technique provides a new and convenient way to characterize the porosity level of porous nanostructures, as well as intrinsic thermal conductivity.

### 4.3. Nano-Crystalline Structure Evolution under Heating

Heat treatment facilitates the transformation between amorphous and crystalline structures. It is hard to observe this kind of structural transformation with conventional imaging methods. Thermal properties can again be a good indicator showing the variation in the state of the crystalline structure because the amorphous and crystalline structures of the same material have differences in their thermal conductivities. Xu et al. [36] applied the PT method to study internal structure transformations of spider silk proteins under heat treatment based on thermal effusivity. Two spider silk protein films prepared from two different types of spiders, *N. clavipes* and *L. Hesperus*, were studied, as shown in Figure 4. When elevating the heating temperature, the thermal effusivity of the protein films significantly increased because of the transformation from random coils (amorphous structure) to α-helices and antiparallel β-sheets (crystalline structure). Supplementary Raman studies of the films showed that the characteristic peak of protein started to shift when the heating temperature reached 60 °C. In the heating process in this low temperature range (lower than 60 °C), the increase in crystallinity was the main reason accounting for the increase in the thermal effusivity. As the temperature increased to more than 80 °C, the Raman characteristic peaks disappeared because the crystalline structures were destroyed due to H-bond breaking among molecular chains. Increases in both thermal conductivity (fewer boundaries) and volumetric heat capacity quickened the increasing rate of the thermal effusivity from 100 to 120 °C. In this work, the Raman spectra of the protein films were strongly affected by fluorescence induced by surface carbonization, especially in the high temperature range, while the thermal effusivity from the PT technique continuously responded well to the structure variation across the whole temperature range. Thus, PT technique could be a good candidate for nanostructure investigation when conventional methods are not applicable.

### 4.4. Considerations in the Measurement of Micro-/Nanomaterials

The abovementioned PT technique is able to measure the cross-plane thermal conductivity, heat capacity (*ρc*_p_), and thermal contact resistance for a multilayered sample under the assumption of 1D heat conductance along the thickness direction. From Equations (5)–(8), it can be seen that the method determines the absolute value of thermal resistance for conductance (the sum of *L*/*κ* and *R*) and heat capacitance (*Lρc*_p_). For a certain layer *i* with a thin thickness *L_i_*, when its thermal resistance (*L_i_*/*κ_i_*) is much smaller than the uncertainty Δ*R* of *R*, *R* will dominate the variation in the PT signal and the change in *L_i_*/*κ_i_* will not be sensed. The minimum thickness in the PT method should thus be larger than Δ*R·κ_i_.* As alternatives, the TDTR and FDTR methods employ an ultrafast pulsed laser (femtosecond to nanosecond) to realize a nanometer-level thermal penetration depth and, thus, the thermal measurement of nanometer-thick coatings [14]. Raman-based thermal methods measure temperature according to the variation in the materials’ characteristic peaks in the Raman spectrum. They are available for both suspended and supported films and need no metal coating on the top of samples [24,26,30,37,38].

Another concern about the PT method is the in-plane thermal conductivity. As described in the physical model, the heating laser spot size should be larger than the in-plane thermal diffusion length in each layer so that the 1D model is valid. However, if the sample has a high in-plane thermal conductivity (such as graphene, etc.), the 1D model is violated. A direct solution is to build a 3D heat conduction model and also consider the spatial distribution of the heating laser [39,40]. For suspended samples, the evaluation of the transient term in Equation (2) in the time domain and the steady-state temperature field mapping can be used achieve in-plane thermal conductivity measurement using the current experimental setup. Moreover, other methods, such as the TDTR [41], FDTR [14], and Raman-based thermal methods [25,42,43], can achieve both kinds of in-plane thermal conductivity measurement.

Furthermore, infrared thermal radiation is not material-specific. When measuring an individual nanostructure, such as a single nanoparticle, an IR detector may gather the thermal radiation from the nanoparticle and its surroundings/supporting materials. The determined thermal properties are thus averages one rather than the properties for a specific nanostructure. In contrast, Raman spectroscopy has a fingerprint feature and can respond to temperature changes and detect the temperature rise for individual micro-/nanostructures [24,38,44,45,46,47].

## 5. Conclusions

In this paper, we reviewed the physical mechanism of PT technique for thermal property measurement and its practical application in the thermal characterization of nanomaterials. With PT technique, the phase shift method provides a high sensitivity to the thermal properties while the amplitude method can measure thermal conductivity without considering interfacial contact. Utilizing the dependency of thermal properties on the internal structure of materials, PT technique has shown its unique capabilities for nanostructure investigation where commonly used micro-/nanoscale imaging technologies might not be applicable. Though limitations exist, PT technique is quite mature. Future application of PT technique can be extended to thermal property and structure detection beneath the surface.

## Figures and Tables

**Figure 1 nanomaterials-12-01884-f001:**
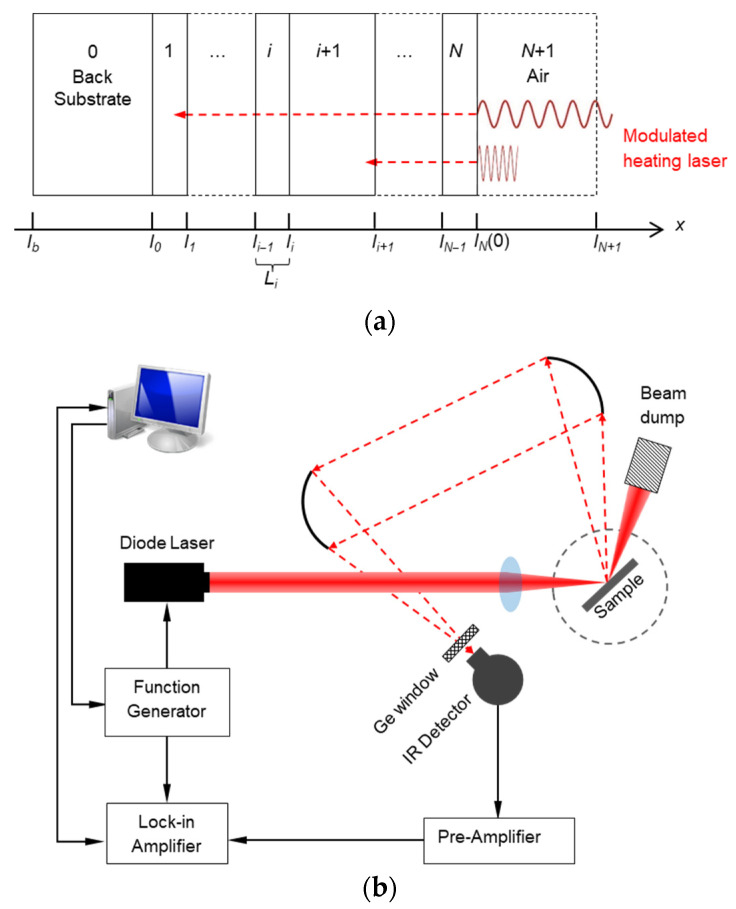
Physical schematics of PT technique: (**a**) mechanism of modulated heating and 1D heat conduction across the multilayered structure; (**b**) a typical experimental setup for PT technique.

**Figure 2 nanomaterials-12-01884-f002:**
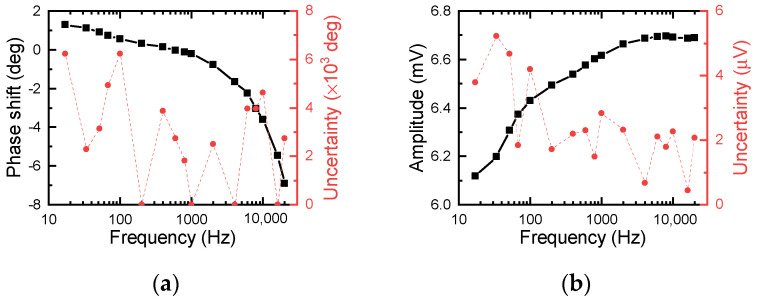
The calibration of a typical experimental setup for PT technique: (**a**) phase shift and (**b**) amplitude. The black square is the measured phase shift and amplitude, and the red dot denotes the measurement uncertainty of the phase shift and amplitude.

**Figure 3 nanomaterials-12-01884-f003:**
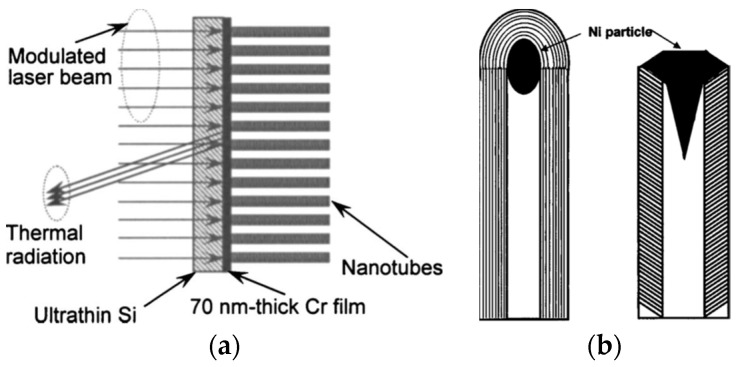
Thermal and structural characterization of CNTs: (**a**) multilayered structure of CNTs; (**b**) schematics of CNTs’ wall growth on a Ni particle with a special structure. Reprinted with permission from Ref. [32], Copyright (2022), AIP Publishing.

**Figure 4 nanomaterials-12-01884-f004:**
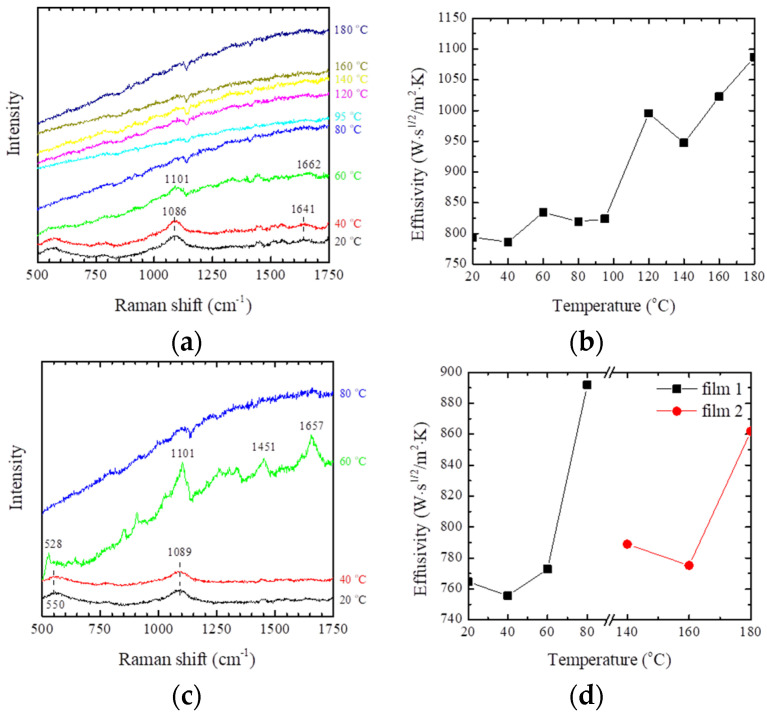
Thermal and structural characterization of spider silk protein film: (**a**) Raman spectra and (**b**) PT determined thermal effusivity of *N. clavipes* spider silk protein film; (**c**) Raman spectra and (**d**) PT determined thermal effusivity of *L. Hesperus* spider silk protein film. Reprinted with permission from Ref. [36], Copyright (2022), Elsevier.

## Data Availability

Not applicable.

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
