# Peer review of "Review of Photothermal Technique for Thermal Measurement of Micro-/Nanomaterials"

_nanomaterials, 2022, doi:10.3390/nano12111884_

Round 1

Reviewer 1 Report

Authors nicely describe the photothermal method to measure micro-and nanoscale materials. They explain background, theory, experimental setup and some applications. However, this review should be completed with more details (mentioned in my comments below) before it is suitable for publication.

-Line 63: Typos: developed by from? Proposed by by..

-Equation 13 gives the temperature distribution in the gas. However, for the photothermal experiment, the radiation emitted by the surface is measured to determined the thermal properties. Therefore, the temperature at lN is the important quantity. Authors should clarify how this temperature is extracted from the gas layer.

-It seems that the theory exposed in the manuscript is used to characterize the thermal properties of multilayered materials. However, it does not seem possible to extract that properties of bulk samples as they have constant phase shift. Many micro- and nanomaterials are fabricated in a bulk manner (pellet, nanostructured bulk, etc). Could these materials be measured? Would one use the amplitude instead of the phase shift? Authors should clarify this point.

-Line 135 is confusing, rewording is needed.

-What do authors mean with “the number of the needed frequency” in line 144? I recommend it to be replaced by “the number of frequency steps” or similar.

-In line 155, when authors speak about calibration, it is not very clearly explained why the Ge window must be removed. Why is the intensity mentioned? Does this experiment need to use any other type of filter then? The sentence is confusing and should be reworded.

-In line 157, authors should explain what Acal (calibration amplitude) is because it was not introduced previously.

-Line 183, the word transparency should be replaced by “transparent”

-Line 200, replace “demstrating” by “demonstrating”

-In Line 255, authors claim that PT is not suitable to measure nanostructures due to the diffraction limit of visible light. However, if the excitation wavelength of the laser is reduced down to 488nm (as an example), then by using an objective, the beam size can be reduced down to 500nm and  can be used to probe single structures. Authors should elaborate a bit more on the reason why it cannot be used for single nanostructures. Moreover, as it is written it seems that it is limited by the wavelength of the thermal radiation, while I believe that the limiting factor is the exciting wavelength. Please, clarify.

-Authors should rework a bit the sentence written in line 260-263. I believe I understand what they mean but it can be improved quite a bit.

-For this experiment, samples are typically coated with a metal layer to absorb the light and generate heat. Authors should comment on this.

-This technique can also be performed on rear configuration, where the thermal emission from the rear surface is detected by the IR detector. Authors should mention this to complete this review.

-Authors should comment on the uncertainty of the technique.

-Authors should comment on improvements/variations of this technique. For example, how this technique has been extended to also measure bulk samples, or even in-plane thermal diffusivity of free-standing samples. Authors can find relevant references in the following review: Abad et al. Non-contact methods for thermal properties measurement, Renewable and sustainable energy reviews, 76 (2017) 1348-1370.

Reviewer 2 Report

The proposed Review work “Review on photothermal technique for thermal measurement of micro-/nanomaterials” aims at introducing the  photothermal (PT) technique as a valid method for the micro/nanoscale solid samples.

Although, the Introduction section reports the investigation of the traditional thermal measurements methods, the scientific advantages of the PT are poorly understood.  I suggest the authors to go into detail of the advantages of PT with respect the conventional methods in order to, also, increase the bibliography, that seems very poor for a review work.

The case studies reported are scientifically interesting as potential applications of the proposed technique. However, I think that it could be more interesting to show the limits of the PT method by reporting possible solutions by literature.

Round 2

Reviewer 1 Report

Thank you for addressing my comments

Reviewer 2 Report

The paper was  properly improved according to the reviewer suggestions. The manuscript is acceptable for publication.